# Heteroskedastic and Imbalanced Deep Learning with Adaptive Regularization

**Kaidi Cao**[1]**, Yining Chen**[1]**, Junwei Lu**[2]**, Nikos Arechiga**[3]**, Adrien Gaidon**[3]**, Tengyu Ma**[1]

[1]Stanford University, [2]Harvard University, [3]Toyota Research Institute
{kaidicao,cynnjjs,tengyuma}@stanford.edu

## Abstract

Real-world large-scale datasets are heteroskedastic and imbalanced – labels have varying levels of uncertainty and label distributions are long-tailed. Heteroskedasticity and imbalance challenge deep learning algorithms due to the difficulty of distinguishing among mislabeled, ambiguous, and rare examples. Addressing heteroskedasticity and imbalance simultaneously is under-explored. We propose a data-dependent regularization technique for heteroskedastic datasets that regularizes different regions of the input space differently. Inspired by the theoretical derivation of the optimal regularization strength in a one-dimensional nonparametric classification setting, our approach adaptively regularizes the data points in higher-uncertainty, lower-density regions more heavily. We test our method on several benchmark tasks, including a real-world heteroskedastic and imbalanced dataset, WebVision. Our experiments corroborate our theory and demonstrate a significant improvement over other methods in noise-robust deep learning.[1]

## 1 Introduction

In real-world machine learning applications, even well-curated training datasets have various types of heterogeneity. Two main types of heterogeneity are: (1) data imbalance: the input or label distribution often has a long-tailed density, and (2) heteroskedasticity: the labels given inputs have varying levels of uncertainties across subsets of data stemming from various sources such as the intrinsic ambiguity of the data or annotation errors. Many deep learning algorithms have been proposed for imbalanced datasets (e.g., see (Wang et al., 2017; Cao et al., 2019; Cui et al., 2019; Liu et al., 2019) and the reference therein). However, heteroskedasticity, a classical notion studied extensively in the statistical community (Pintore et al., 2006; Wang et al., 2013; Tibshirani et al., 2014), has so far been under-explored in deep learning. This paper focuses on addressing heteroskedasticity and its interaction with data imbalance in deep learning.

Heteroskedasticity is often studied in regression analysis and refers to the property that the distribution of the error varies across inputs. In this work, we mostly focus on classification, though the developed technique also applies to regression. Here, heteroskedasticity reflects how the uncertainty in the conditional distribution $y \mid x$, or the entropy of $y \mid x$, varies as a function of $x$. Real-world datasets are often heteroskedastic. For example, Li et al. (2017) shows that the WebVision dataset has a varying number of ambiguous or true noisy examples across classes.[2]

Conversely, we consider a dataset to be *homoscedastic* if every example is mislabeled with a fixed probably $\epsilon$, as assumed by many prior theoretical and empirical works on label corruption (Ghosh et al., 2017; Han et al., 2018; Jiang et al., 2018; Mirzasoleiman et al., 2020). We note that varying uncertainty in $y \mid x$ can come from at least two sources: the intrinsic semantic ambiguity of the input, and the (data-dependent) mislabeling introduced by the annotation process. Our approach can handle both types of noisy examples in a unified way, but for the sake of comparisons with past methods, we call them "ambiguous examples" and "mislabeled examples" respectively, and refer to both of them as "noisy examples".

---

[1]Code available at https://github.com/kaidic/HAR.

[2]See Figure 4 of (Li et al., 2017), the number of votes for each example indicates the level of uncertainty of that example.

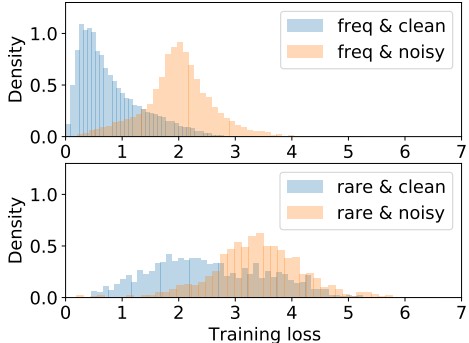

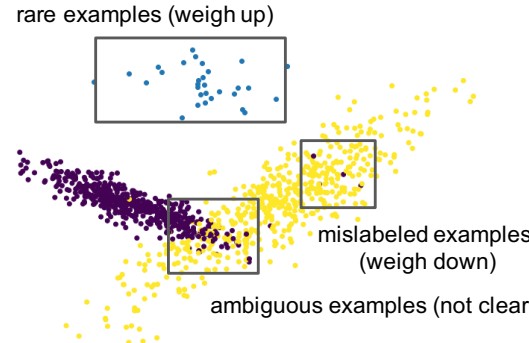

rare examples (weigh up)

mislabeled examples
(weigh down)

ambiguous examples (not clear)

Figure 1: Histogram of the distributions of losses on an imbalanced and noisy CIFAR-10 dataset. Clean but rare examples tend to have larger losses, similar to the noisy examples in frequent classes.

Figure 2: Real-world datasets have various sources of heterogeneity and it could be hard to distinguish one from another. They require mutually-exclusive reweighting strategy, but they all benefit from stronger regularization.

Overparameterized deep learning models tend to overfit more to the noisy examples (Arpit et al., 2017; Frénay & Verleysen, 2013; Zhang et al., 2016). To address this issue, a common approach is to detect noisy examples by selecting those with large training losses, and then remove them from the (re-)training process. However, an input's training loss can also be big because it is rare or ambiguous (Hacohen & Weinshall, 2019; Wang et al., 2019), as shown in Figure 1. Noise-cleaning methods could fail to distinguish mislabeled from rare/ambiguous examples (see Section 3.1 for empirical proofs). Though dropping the former is desirable, dropping the latter loses important information. Another popular approach is reweighting methods that reduce the contribution of noisy examples in optimization. However, failing to distinguish between mislabeled and rare/ambiguous examples makes the decision of the weights tricky – mislabeled examples require small weights, whereas rare / ambiguous examples benefit from larger weights (Cao et al., 2019; Shu et al., 2019).

We propose a regularization method that deals with noisy and rare examples in a unified way. We observe that mislabeled, ambiguous, and rare examples all benefit from stronger regularization (Hu et al., 2020; Cao et al., 2019). We apply a Lipschitz regularizer (Wei & Ma, 2019a;b) with varying regularization strength depending on the particular data point. Through theoretical analysis in the one-dimensional setting, we derive the *optimal* regularization strength for each training example. The optimal strength is larger for rarer and noisier examples. Our proposed algorithm, heteroskedastic adaptive regularization (HAR), first estimates the noise level and density of each example, and then optimizes a Lipschitz-regularized objective with input-dependent regularization with strength provided by the theoretical formula.

In summary, our main contributions are: (i) we propose to learn heteroskedastic imbalanced datasets under a unified framework, and theoretically study the optimal regularization strength on one-dimensional data. (ii) we propose an algorithm, heteroskedastic adaptive regularization (HAR), which applies stronger regularization to data points with high uncertainty and low density. (iii) we experimentally show that HAR achieves significant improvements over other noise-robust deep learning methods on simulated vision and language datasets with controllable degrees of data noise and data imbalance, as well as a real-world heteroskedastic and imbalanced dataset, WebVision.

## 2 ADAPTIVE REGULARIZATION FOR HETEROSKEDASTIC DATASETS

### 2.1 BACKGROUNDS

We first introduce general nonparametric tools that we use in our analysis, and review the dependency of optimal regularization strength on the sample size and noise level.

**Over-parameterized neural networks as nonparametric methods.** We use nonparametric method as a surrogate for neural networks because they have been shown to be closely related. Recent work (Savarese et al., 2019) shows that the minimum norm two-layer ReLU network that fits

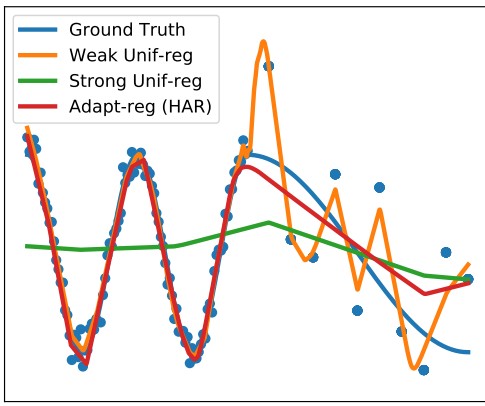

Figure 3: A one-dimensional example with a three-layer neural network in heteroskedastic and imbalanced regression setting. The curve in blue is the underlying ground truth and the dots are observations with heteroskedastic noise. This example shows that uniformly weak regularization overfits on noisy and rare data (on the right half), whereas uniformly strong regularization causes underfitting on the frequent and oscillating data (on the left half). The adaptive regularization does not underfit the oscillating data but still denoise the noisy data. We note that standard nonparametric methods such as cubic spline do not work here because they also use uniform regularization.

the training data is in fact a linear spline interpolation. Parhi & Nowak (2019) extend this result to a broader family of neural networks with a broader family of activations.

Given a training dataset $\{(x_i, y_i)\}_{i=1}^n$, nonparametric method with penalty works as follows. Let $\mathcal{F} : \mathbb{R} \to \mathbb{R}$ be a twice-differentiable model family. We aim to fit the data with smoothness penalty

$$\min_f \ \frac{1}{n} \sum_{i=1}^n \ell(f(x_i), y_i) + \lambda \int (f'(x))^2 dx \qquad (1)$$

**Lipschitz regularization for neural networks.** Lipschitz regularization has been shown to be effective for deep neural networks as well. Wei & Ma (2019a) proves a generalization bound of neural networks dependent on the Lipschitzness of each layer with respect to all intermediate layers on the training data, and show that, empirically, regularizing the Lipschitzness improve the generalization. Sokolić et al. (2017) shows similar results in data-limited settings. In Section 2.3, we extend the Lipschitz regularization technique to heteroskedastic setting.

**Regularization strength as a function of noise level and sample size.** Finally, we briefly review existing theoretical insights on the optimal choice of regularization strength. Generally, the optimal regularization strength for a given model family increases with the label noise level and decreases in the sample size. As a simple example, consider linear ridge regression $\min_\theta \frac{1}{n} \sum_{i=1}^n (x_i^\top \theta - y_i)^2 + \lambda \|\theta\|^2$, where $x_i, \theta \in \mathbb{R}^d$ and $y_i \in \mathbb{R}$. We assume $y_i = x_i^\top \theta^* + \xi$ for some ground truth parameter $\theta^*$, and $\xi \sim \mathcal{N}(0, \sigma^2)$. Then the optimal regularization strength $\lambda_{opt} = d\sigma^2/n\|\theta^*\|_2^2$. Results of similar nature can also be found in nonparametric statistics (Wang et al., 2013; Tibshirani et al., 2014).

## 2.2 Heteroskedastic Nonparametric Classification on One-Dimensional Data

We consider a one-dimensional binary classification problem where $\mathcal{X} = [0, 1] \subset \mathbb{R}$ and $\mathcal{Y} = \{-1, 1\}$. We assume $Y$ given $X$ follows a logistic model with ground-truth function $f^\star$, i.e.

$$\Pr[Y = y | X = x] = \frac{1}{1 + \exp(-yf^\star(x))}. \qquad (2)$$

The training objective is cross-entropy loss plus Lipschitz regularization, i.e.

$$\hat{f} = \operatorname{argmin}_f \ \widehat{L}(f) \triangleq \frac{1}{n} \sum_{i=1}^n \ell(f(x_i), y_i) + \lambda \int_0^1 \rho(x)(f'(x))^2 dx, \qquad (3)$$

where $\ell(a, y) = -\log(1 + \exp(-ya))$, and $\rho(x)$ is a smoothing parameter as a function of the noise level and density of $x$. Let $I(x)$ be the fisher information matrix conditioned on the input, i.e. $I(x) \triangleq \mathbb{E}[\nabla_a^2 \ell(a, Y)|_{a=f^\star(X)} | X = x]$. When $(X, Y)$ follows the logistic model in equation 2,

$$I(x) = \frac{1}{(1 + \exp(f^\star(x))(1 + \exp(-f^\star(x))} = \operatorname{Var}(Y | X = x).$$

Therefore, $I(x)$ captures the aleatoric uncertainty of $x$. For example, when $Y$ is deterministic conditioned on $X = x$, we have $I(x) = 0$, indicating perfect certainty.

Define the test metric as the mean-squared-error on the test set $\{(x_i, y_i)\}_{i=1}^n$, i.e.,[1]

$$\text{MSE}(\hat{f}) \triangleq \mathop{\mathbb{E}}_{\{(x_i, y_i)\}_{i=1}^n} \int_0^1 (\hat{f}(t) - f^\star(t))^2 dt \tag{4}$$

Our main goal is to derive the optimal choice of $\rho(x)$ that minimizes the MSE. We start with an analytical characterization of the test error. Let $W_2^2 = \{f' \text{ is absolute continuous and } f'' \in L^2[0,1]\}$. We denote the density of $X$ as $q(x)$. The following theorem analytically computes the MSE under the regularization strength $\rho(\cdot)$, building upon (Wang et al., 2013) for regression problems. The proof of the Theorem is deferred to Appendix A.

**Theorem 1.** *Assume that $f^\star, q, I \in W_2^2$. Let $r(t) = -1/(q(t)I(t))$ and $L_0 = \int_{-\infty}^\infty \frac{1}{4}\exp(-2|t|)dt$. If we choose $\lambda = C_0 n^{-2/5}$ for some constant $C_0 > 0$, the asymptotic mean squared error is*

$$\lim_{n\to\infty} MSE(\hat{f}) = C_n \int_0^1 \lambda^2 r^2(t) \left[\frac{d}{dt}(\rho(t)(f^*)'(t))\right]^2 + L_0 r(t)^{1/2}\rho(t)^{-1/2} dt$$

*in probability, where $C_n$ is a scalar that only depends on $n$.*

Using the analytical formula of the test error above, we want to derive an approximately optimal choice of $\rho(x)$. A precise computation is infeasible, so we restrict ourselves to consider only $\rho(x)$ that is constant within groups of examples. We introduce an additional structure – we assume the data can be divided into $k$ groups $[a_0, a_1), [a_1, a_2), \cdots, [a_{k-1}, a_k)$. Each group $[a_j, a_{j+1})$ consists of an interval of data with approximately the same aleatoric uncertainty. We approximate $\rho(t)$ is constant on each of the group $[a_i, a_{i+1})$ with value $\rho_i$. Plugging this piece-wise constant $\rho$ into the asymptotic MSE in Theorem 1, we obtain

$$\lim_{n\to\infty} \text{MSE}(\hat{f}) = \sum_j \left[\rho_j^2 \int_{a_j}^{a_{j+1}} r^2(t) \left[\frac{d^2}{dt^2}f^\star(t)\right]^2 dt + \rho_j^{-1/2} L_0 \int_{a_j}^{a_{j+1}} r^{1/2}(t)dt\right].$$

Minimizing the above formula over $\rho_1, \ldots, \rho_k$ separately, we derive the optimal weights, $\rho_j = \left[\dfrac{L_0 \int_{a_j}^{a_{j+1}} r(t)^{1/2}dt}{4 \int_{a_j}^{a_{j+1}} r^2(t)\left[\frac{d^2}{dt^2}f^\star(t)\right]^2 dt}\right]^{2/5}$. In practice, we do not know $f^\star$ and $q(x)$, so we make the following simplifications. We assume that $q(t)$ and $I(t)$ are constant on each interval $[a_j, a_{j+1}]$. In other words, we assume that $q(t) = q_j$ and $I(t) = I_j$ for all $t \in [a_j, a_{j+1}]$. We further assume that $\frac{d^2}{dt^2}f^\star(t)$ is close to a constant on the entire space, because estimating the curvature in high dimension is difficult. This simplification yields $\rho_j \propto \left[\dfrac{q_j^{-1/2}I_j^{-1/2}}{q_j^{-2}I_j^{-2}}\right]^{2/5} = q_j^{3/5}I_j^{3/5}$. We find the simplification works well in practice.

**Adaptive regularization with importance sampling.** It is practically infeasible to implement the integration in equation 3 for high-dimensional data. We use importance sampling to approximate the integral:

$$\text{minimize}_f \quad L(f) \triangleq \frac{1}{n}\sum_{i=1}^n \ell(f(x_i), y_i) + \lambda \sum_{i=1}^n \tau_i f'(x_i)^2 \tag{5}$$

Suppose $x_i \in [a_j, a_{j+1})$, we have that $\tau_i$ should satisfy that $\tau_i q_j = \rho_j$ so that the expectation of the regularization term in equation 5 is equal to that in equation 3. Hence,

$$\tau_i = I_j^{3/5}q_j^{-2/5} = I(x_i)^{3/5}q(x_i)^{-2/5}.$$

**Adaptive regularization for multi-class classification and regression.** In fact, the proof of Theorem 1 is proved for general loss $\ell(a, y)$. Therefore, we can directly generalize it to multi-class classification and regression problems. For a regression problem, $\ell(a, y)$ is the square loss: $\ell(y, a) = 0.5(y - a)^2$, the Fisher information $I(x) = 1$. Therefore, for a regression problem, we can choose regularization weight $\tau_i = q(x_i)^{-2/5}$.

---

[1]Note that we integrate the error without weighting because we are interested in the balanced test performance.

## 2.3 PRACTICAL IMPLEMENTATION ON NEURAL NETWORKS WITH HIGH-DIMENSIONAL DATA

We heuristically extend the Lipschitz regularization technique discussed in Section 2.2 from non-parametric models to over-parameterized deep neural networks. Let $(x, y)$ be an example and $f_\theta$ be an $r$-layer neural network. We denote by $h^{(j)}$ the $j$-th hidden layer of the network, by $J^{(j)}(x) \triangleq \frac{\partial}{\partial h^{(j)}} \mathcal{L}(f(x), y)$, i.e., the Jacobian of the loss w.r.t $h^{(j)}$. We replace the regularization term $f'(x)^2$ in equation 5 by $R(x) = \left( \sum_{j=1}^r \|J^{(j)}(x)\|_F^2 \right)^{1/2}$, which was proposed by (Wei & Ma, 2019a). As a proof of concept, we visualize the behavior of our algorithm in Figure 3, where we observe that the rare and noisy examples have significantly improved error due to stronger regularization. In contrast, a uniform regularization either overfits or underfits different subsets.

Note that the differences from the 1-D case include the following three aspects. 1. The derivative is taken w.r.t to all the hidden layers for deep models, which has been shown to have superior generalization guarantees for neural networks by (Wei & Ma, 2019a;b). 2. An additional square root is taken in computing $R(x)$. This modified version may have milder curvature and be easier to tune. 3. We take the derivative of the loss instead of the derivative of the model, which outputs $k$ numbers for multi-class classification. This is because the derivative of the model requires $k$ times more time to compute. The regularized training objective is consequently

$$\text{minimize}_f \quad L(f) \triangleq \frac{1}{n} \sum_{i=1}^n \left( \ell(f(x_i), y_i) + \lambda \tau_i R(x_i) \right), \tag{6}$$

where $\tau_i$ is chosen to be $\tau_i = I(x_i)^{3/5}/q(x_i)^{2/5}$ following the formula equation 5 in Section 2.2 and $\lambda$ is a hyperparameter to control the overall scale of the regularization strength. We note that we do not expect this choice of $\tau_i$ to be optimal for the high-dimensional case with all the modifications above – the optimal choice does depend on the nuances. However, we also observe that the empirical performance is not sensitive to the form of $\tau$ as long as it's increasing in $I(x)$ and decreasing in $q(x)$. That is, the more uncertain or rare an example is, the stronger regularization should be applied.

In order to estimate the relative regularization strength $\tau_i$, the key difficulty lies in the estimation of uncertainty $I(x)$. As in the 1-D setting, we divide the examples into $k$ groups $G_1, \ldots, G_k$ (e.g., each group can correspond to a class), and estimate the uncertainty on each group. In the binary setting, $I(x) = \text{Var}(Y|X = x) = \Pr[Y = 1 \mid X] \cdot \Pr[Y = 0 \mid X]$ can be approximated by $\tilde{I}(x) = 1 - \max_{k \in \{0,1\}} \Pr[Y = k \mid X = x]$ up to a factor of at most 2. We use the same formula for multi-class setting as the approximation of the uncertainty. (As a sanity check, when $Y$ is concentrated on a single outcome, the uncertainty is 0.) Note that $\tilde{I}(x)$ is essentially the minimum possible error of any deterministic prediction on the data point $x$. Assume that we have a sufficiently accurate pre-trained model, we can use its validation error to estimate $\tilde{I}(x)$:

Then for all $x \in G_j$, we estimate $q(x)$ and $I(x)$ by

$$\forall x \in G_j, q(x) \propto |G_j|, \ I(x) \propto \text{average validation error of a pre-trained model } f_{\tilde{\theta}} \text{ on } G_j \tag{7}$$

The whole training pipeline is summarized in Algorithm 1.

---

**Algorithm 1** Heteroskedastic Adaptive Regularization (HAR)

---

**Require:** Dataset $\mathcal{D} = \{(x_i, y_i)\}_{i=1}^n$. A parameterized model $f_\theta$
 1: Split training set $\mathcal{D}$ into $\mathcal{D}_{\text{train}}$ and $\mathcal{D}_{\text{val}}$
 2: $f_{\tilde{\theta}} \leftarrow$ Standard SGD Training on $\mathcal{D}_{\text{train}}$
 3: Estimate $I(x), q(x)$ with equation 7 using $f_{\tilde{\theta}}$ on $\mathcal{D}_{\text{val}}$, and compute $\tau_i = I(x_i)^{3/5}/q(x_i)^{2/5}$
 4:
 5: Initialize the model parameters $\theta$ randomly
 6: $f_\theta \leftarrow$ SGD with the regularized objective as in equation 6 on the full dataset $\mathcal{D}$

---

## 3 EXPERIMENTS

We experimentally show that our proposed algorithm HAR(Algorithm 1) improves the test performance of the noisier and rarer groups of examples (by stronger regularization) without negatively

affecting the training and test performance of the other groups. We evaluate our algorithms on three vision datasets and one NLP dataset: CIFAR-10 and CIFAR-100 (Krizhevsky et al., 2009), IMDB-review (Maas et al., 2011) (see Appendix C.1), and WebVision (Li et al., 2017), a real-world heteroskedastic and imbalanced dataset. Please refer to Appendix B for low-level implementation details.

**Baselines.** We compare our proposed HAR with the following baselines. The simplest one is (1) Empirical risk minimization (ERM): the vanilla cross-entropy loss with all examples having the same weights of losses. We select two representatives from the **noise-cleaning** line of approach. (2) Co-teaching (Han et al., 2018): two deep networks are trained simultaneously. Each network aims to identify clean data points that have small losses and use them to guide the training of the other network. (3) INCV (Chen et al., 2019): it extends Co-teacing to an interative version to estimate the noise ratio and select data. We consider three representatives from the **reweighting-based methods**, including two that learn the weighting using **meta-learning**. (4) MentorNet (Jiang et al., 2018): it pretrains a teacher network that outputs weights for examples that are used to train the student network with reweighting. (5) L2RW (Ren et al., 2018): it directly optimizes weights of each example in the training set by minimizing its corresponding loss on a small meta validation set. (6) MW-Net (Shu et al., 2019): it extends L2RW by explicitly defining a weighting function which depends only on the loss of the example. We also compare against two representatives from the **robust loss function**. (7) GCE (Zhang & Sabuncu, 2018): it generalizes mean average error and cross-entropy loss to obtain a new loss function. (8) DMI (Xu et al., 2019): it designs a new loss function based on generalized mutual information. In addition, as an essential ablation study, we consider vanilla uniform regularization. (9) Unif-reg: we apply the Jacobian regularizer on all examples with equal strength, and tune the strength to get the best possible validation accuracy.

### 3.1 SIMULATING HETEROSKEDASTIC AND IMBALANCED DATASETS ON CIFAR

**Setting.** Unlike previous works that test on uniform random or asymmetric noise, which is often not the case in reality, in this paper we test our method on more realistic noisy settings, as suggested by Patrini et al. (2017); Zhang & Sabuncu (2018). In order to simulate heteroskedasticity, we only corrupt semantically-similar classes. For CIFAR-10, we exchange 40% of the labels between classes 'cat' and 'dog', and between 'truck' and 'automobile'. CIFAR-100 has 100 classes grouped into 20 super classes. For each class of the 5 classes under the super class 'vehicles 1' and 'vehicles 2', we corrupt the labels with 40% probability uniformly randomly to the rest of four classes under the same super class. As a result, the 10 classes under super class 'vehicle 1' and 'vehicle 2' have high label noise level and the corruption are only within the same super class. Heteroskedasticity of the labels and imbalance of the inputs commonly coexist in the real world settings. HAR can take both of them into account. To understand the challenge imposed by the entanglements of heteroskedasticity and imbalance, and compare HAR with the aforementioned baselines, we inject data imbalance concurrently with the heteroskedastic noise. We remove samples from the corrupted classes to simulate the most difficult scenario — the rare and noisy groups overfit significantly. (A more benign interaction between the noises and imbalance is that the rare classes have lower noise level, we defer it to Appendix C.3.) We use the imbalance ratio to denote the frequency ratio between the frequent (and clean) classes to the rare (and corrupted) classes. We consider imbalance ratio to be 10 and 100.

**Result.** Table 1 summarizes the results. Since examples from rare classes tend to have larger training and validation loss regardless of whether the labels are correct or not, noise-cleaning based methods might drop excessive examples with correct labels. We examined the noise ratio of dropped samples for INCV under the setting of imbalance ratio equals 10. Among all dropped examples, there is only 19.2% of true noise examples. In addition, the rare class examples selected still have 29.8% of label noise. This explains that the significant decrease of accuracies of Co-teaching and INCV on corrupted and rare classes. Reweighting-based methods tend to suffer from the loss of accuracy in other more frequent classes, which is aligned with the findings in Cao et al. (2019). While the aforementioned baselines struggle to deal with heteroskedasticity and imbalance together, HAR is able to put them under the same regularization framework and achieve significant improvements. Notably, HAR also shows improvement over uniform regularization with optimally tuned strength. This clearly demonstrates the importance of introducing adaptive regularization among all examples for a better trade-off. A more detailed ablation study on the trade-off between training accuracy and validation accuracy can be found in Section 3.3.

Table 1: Top-1 validation accuracy (averaged over 3 runs) of ResNet-32 on heteroskedastic and imbalanced CIFAR-10. HAR significantly improves noisy and rare classes, while keeping the accuracy on other classes almost unaffected.

| Imbalance ratio | 10 | | 100 | |
| Method | Noisy&Rare Cls. | Clean Cls. | Noisy&Rare Cls. | Clean Cls. |
|---|---|---|---|---|
| ERM | $52.9 \pm 1.2$ | $94.4 \pm 0.1$ | $18.9 \pm 1.0$ | $94.2 \pm 0.1$ |
| Co-teaching | $30.2 \pm 2.3$ | $88.9 \pm 0.3$ | $15.4 \pm 2.8$ | $86.4 \pm 0.7$ |
| INCV | $48.9 \pm 1.7$ | $94.0 \pm 0.2$ | $25.8 \pm 1.8$ | $93.8 \pm 0.2$ |
| MentorNet | $54.1 \pm 1.0$ | $90.3 \pm 0.5$ | $28.3 \pm 1.5$ | $90.2 \pm 0.4$ |
| L2RW | $44.3 \pm 2.0$ | $90.1 \pm 0.5$ | $31.2 \pm 1.9$ | $89.7 \pm 0.7$ |
| MW-Net | $55.4 \pm 1.1$ | $91.7 \pm 0.5$ | $35.6 \pm 1.6$ | $92.3 \pm 0.5$ |
| GCE | $48.2 \pm 0.6$ | $91.6 \pm 0.3$ | $14.1 \pm 2.0$ | $91.7 \pm 0.4$ |
| DMI | $44.7 \pm 2.3$ | $90.7 \pm 0.8$ | $14.0 \pm 2.1$ | $91.8 \pm 0.6$ |
| Unif-reg (optimal) | $53.9 \pm 0.9$ | $92.1 \pm 0.2$ | $36.7 \pm 1.0$ | $92.4 \pm 0.3$ |
| **Ours (HAR)** | $\mathbf{63.5 \pm 0.8}$ | $\mathbf{94.3 \pm 0.2}$ | $\mathbf{42.4 \pm 0.7}$ | $\mathbf{94.0 \pm 0.2}$ |

## 3.2 ABLATION STUDY ON CIFAR

We disentangle the problem setting to show the effectiveness of our unified framework.

**Simulating heteroskedastic noise on CIFAR.** We study the uncertainty part of HAR by testing under the setting with only heteroskedastic noise. The type of noise injection is the same as Section 3.1.

We report the top-1 validation accuracy of various methods in Table 2. Aligned with our analysis in Section 4, we observe that both noise-cleaning and reweighting based methods don't get a comparable accuracy on noisy classes with applying strong regularization ($\lambda = 0.1$) under this heteroskedastic setting. We observe the behavior that too strong regularization impede the model from fitting informative samples, thus it could lead to a decrease on clean classes' accuracy. On the contrary, too weak regularization leads to overfitting the noisy examples thus the accuracy on noisy classes do not reach the optimal.

Interestingly, we find that even the well-studied CIFAR-100 dataset has intrinsic heteroskedasticity and HAR can improve over uniform regularization to some extent. Please refer to Appendix C.2 for the results on CIFAR-100 and Appendix C.1 for results on IMDB-review.

Table 2: Top-1 validation accuracy (averaged over 3 runs) of ResNet-32 on heteroskedastic CIFAR-10 and CIFAR-100 for the noisy classes and the clean classes.

| Dataset | CIFAR-10 | | CIFAR-100 | |
| Method | Avg. Noisy Cls. | Avg. Clean Cls. | Avg. Noisy Cls. | Avg. Clean Cls. |
|---|---|---|---|---|
| ERM | $68.6 \pm 0.2$ | $93.6 \pm 0.2$ | $65.3 \pm 0.3$ | $67.8 \pm 0.2$ |
| Co-teaching | $64.7 \pm 0.4$ | $89.1 \pm 0.3$ | $59.8 \pm 0.4$ | $65.3 \pm 0.3$ |
| INCV | $76.7 \pm 0.6$ | $93.0 \pm 0.2$ | $66.2 \pm 0.3$ | $68.6 \pm 0.2$ |
| MentorNet | $71.1 \pm 0.4$ | $93.7 \pm 0.2$ | $65.9 \pm 0.3$ | $67.5 \pm 0.3$ |
| L2RW | $70.1 \pm 0.3$ | $92.5 \pm 0.3$ | $65.1 \pm 0.5$ | $67.0 \pm 0.3$ |
| MW-Net | $75.0 \pm 0.3$ | $94.4 \pm 0.2$ | $65.7 \pm 0.3$ | $69.1 \pm 0.2$ |
| GCE | $62.6 \pm 1.1$ | $90.2 \pm 0.2$ | $61.2 \pm 0.6$ | $66.9 \pm 0.2$ |
| DMI | $73.2 \pm 0.7$ | $90.8 \pm 0.2$ | $64.8 \pm 0.5$ | $67.1 \pm 0.2$ |
| Unif-reg ($\lambda = 0.1$) | $77.5 \pm 0.6$ | $92.3 \pm 0.2$ | $69.3 \pm 0.5$ | $66.6 \pm 0.3$ |
| Unif-reg (optimal) | $75.3 \pm 0.3$ | $94.1 \pm 0.2$ | $68.5 \pm 0.3$ | $68.6 \pm 0.2$ |
| **Ours (HAR)** | $\mathbf{80.7 \pm 0.3}$ | $\mathbf{94.5 \pm 0.2}$ | $\mathbf{74.2 \pm 0.3}$ | $\mathbf{69.3 \pm 0.2}$ |

**Simulating data imbalance on CIFAR.** We study the density part of HAR by testing under the setting with only data imbalance. We follow the same setting as Cao et al. (2019) to create imbalanced CIFAR. Long-tailed imbalance follows an exponential decay in sample sizes across different classes. For step imbalance setting, all rare classes have the same sample size, as do all frequent classes. Our approach achieves better results than LDAM-DRW and is comparable to recent state-of-the-art methods under the imbalanced setting.

Table 3: Top-1 validation errors of ResNet-32 on imbalanced CIFAR-10 and CIFAR-100.

| Dataset | Imbalanced CIFAR-10 | | | | Imbalanced CIFAR-100 | | | |
|---|---|---|---|---|---|---|---|---|
| Imbalance Type | long-tailed | | step | | long-tailed | | step | |
| Imbalance Ratio | 100 | 10 | 100 | 10 | 100 | 10 | 100 | 10 |
| ERM | 29.64 | 13.61 | 36.70 | 17.50 | 61.68 | 44.30 | 61.45 | 45.37 |
| Focal | 29.62 | 13.34 | 36.09 | 16.36 | 61.59 | 44.22 | 61.43 | 46.54 |
| CB Focal | 25.43 | 12.90 | 39.73 | 16.54 | 63.98 | 42.01 | 80.24 | 49.98 |
| LDAM-DRW | 22.97 | 11.84 | 23.08 | 12.19 | 57.96 | 41.29 | 54.64 | 40.54 |
| BBN (Zhou et al., 2020) | **20.18** | 11.68 | 21.64 | 11.99 | 57.44 | 40.88 | 57.44 | 40.36 |
| **HAR-DRW** | 20.46 | **10.62** | **20.27** | **11.58** | **55.35** | **38.98** | **51.73** | **37.54** |

Table 4: Validation accuracy of ResNet-50 when tuning the regularization strength on mini WebVision. HAR stands out of the trade-off constraint of fitting and generalization.

| | Train Acc | | Val Acc | |
|---|---|---|---|---|
| **Reg Strength** | Top 1 | Top 5 | Top 1 | Top 5 |
| 0 | 69.01 | 88.64 | 59.40 | 80.84 |
| Unif-reg ($\lambda = 0.01$) | 68.96 | 88.54 | 64.32 | 86.11 |
| Unif-reg ($\lambda = 0.02$) | 67.02 | 87.51 | 64.40 | 85.92 |
| Unif-reg ($\lambda = 0.05$) | 65.11 | 86.33 | 65.80 | 86.84 |
| Unif-reg ($\lambda = 0.1$) | 63.35 | 84.98 | 65.04 | 86.56 |
| **Adaptive (HAR)** | 69.12 | 88.41 | **69.20** | **88.96** |

## 3.3 Evaluation on WebVision with real-world heterogeneity

WebVision (Li et al., 2017) contains 2.4 million images crawled from Google and Flickr using 1,000 labels shared with the ImageNet dataset. Its training set is both heteroskedastic and imbalanced (detailed statistics can be found in (Li et al., 2017)), and it is considered as a popular benchmark for noise robust learning. As the full dataset is very large, we follow (Jiang et al., 2018) to use a mini version, which contains the first 50 classes of the Google subset of the data. Following the standard protocol (Jiang et al., 2018), we test the trained model on the WebVision validation set and the ImageNet validation set. We use ResNet-50 for ablation study and InceptionResNet-v2 for a fair comparison with the baselines. We report results comparing against other state-of-the-art approaches in Table 5. Strikingly, HAR achieves significant improvement.

**Ablation study.** We demonstrate the trade-off between training accuracy and validation accuracy on mini WebVision with various uniform regularization strength and HAR in Table 4. It's evident that when we gradually increase the overall uniform regularization strength, the training accuracy continues to decrease, and the validation accuracy reaches its peak at 5e-2. While a strong regularization could improve generalization, it reduces deep networks' capacity to fit the training data. However, with our proposed HAR, we only enforce strong regularization on a subset so that we improve the generalization on noisier groups while maintaining the overall training accuracy not affected.

## 4 Related Work

Our work is closely related to the following methods and directions.

**Noise-cleaning.** The key idea of noise-cleaning is to identify and remove (or re-label) examples with wrong annotations. The general procedure for identifying mislabeled instances has a long history (Brodley & Friedl, 1999; Wilson & Martinez, 1997; Zhao & Nishida, 1995). Some recent works tailored this idea for deep neural networks. Veit et al. (2017) trains a label cleaning network on a small set of data with clean labels, and uses this model to identify noises in large datasets. To circumvent the requirement of a clean subset, Malach & Shalev-Shwartz (2017) train two networks simultaneously and perform update steps only in case of disagreement. Similarly, in co-teaching (Han et al., 2018), each network selects a certain number of small-loss samples and feeds them to its peer network. Chen et al. (2019) further extends the co-training strategy and comes up with an iterative

Table 5: Validation accuracy of InceptionResNet-v2 on WebVision and ImageNet validation sets. HAR demonstrates significant improvements over the previous state-of-the-arts.

| Train | mini WebVision | | | | full WebVision | | | |
|---|---|---|---|---|---|---|---|---|
| Test | WebVision | | ImageNet | | WebVision | | ImageNet | |
| Method | Top 1 | Top 5 | Top 1 | Top 5 | Top 1 | Top 5 | Top 1 | Top 5 |
| ERM | 62.5 | 80.8 | 58.5 | 81.8 | 69.7 | 87.0 | 62.9 | 83.6 |
| Co-teaching | 63.6 | 85.2 | 61.5 | 84.7 | - | - | - | - |
| INCV | 65.2 | 85.3 | 61.6 | 85.0 | - | - | - | - |
| MentorNet | 63.0 | 81.4 | 57.8 | 79.9 | 70.8 | 88.0 | 62.5 | 83.0 |
| **Ours (HAR)** | **75.5** | **90.7** | **70.3** | **90.0** | **75.0** | **90.6** | **67.1** | **86.7** |

version that performs even better empirically. Recently Song et al. (2020) discovers that it is not necessary to maintain two networks. Removing examples whose training loss exceeds a certain threshold before learning rate decay can also get robust performance.

**Reweighting.** Reweighting training data has shown its effectiveness on noisy data (Liu & Tao, 2015). Its challenge lies in the difficulty of weights estimation. Ren et al. (2018) proposes a meta-learning algorithm to assign weights to training examples based on its gradient direction with the one on a clean validation set. Recently, Shu et al. (2019) proposes to learn an explicit loss-weight function to mitigate the optimizing issue of (Ren et al., 2018). Another line of work resorts to curriculum learning by either designing an easy-to-hard strategy of training (Guo et al., 2018) or introducing an extra network (Jiang et al., 2018) to assign weights.

Noise-cleaning and reweighting methods usually rely on the empirical loss to determine if a sample is noisy. However, when the dataset is heteroskedastic, each example's training/validation loss no longer correlates well with its noise level. In such cases, we argue that changing the strength of regularization is a more conservative adaption and suffers less from uncertain estimation, compared to changing the weights of losses (Please refer to Section C.4 for empirical justifications).

**Robust loss function.** Another line of works has attempted to design robust loss functions (Ghosh et al., 2017; Xu et al., 2019; Zhang & Sabuncu, 2018; Patrini et al., 2017; Cheng et al., 2017; Menon et al., 2016). They usually rely on prior assumption about latent transition matrix that might not hold in practice. On the contrary, we focus on more realistic settings.

**Regularization.** Regularization based techniques have also been explored to combat label noise. Li et al. (2019) proves that SGD with early stopping is robust to label noise. Hu et al. (2020) provides theoretical analysis of two additional regularization methods. While these methods consider a uniform regularization on all training examples, our work emphasizes on adjusting the weights of regularizers in search of a better generalization than uniform assignment.

## 5    CONCLUSION

We propose a unified framework (HAR) for training on heteroskedastic and imbalanced datasets. Our method achieves significant improvements over the previous state-of-the-arts on a variety of benchmark vision and language tasks. We provide theoretical results as well as empirical justifications by showing that ambiguous, mislabeled, and rare examples all benefit from stronger regularization. We further provide the formula for optimal weighting of regularization. Heteroskedasticity of datasets is a fascinating direction worth exploring, and it is an important step towards a better understanding of real-world scenarios in the wild.

## ACKNOWLEDGEMENTS

Toyota Research Institute ("TRI") provided funds and computational resources to assist the authors with their research but this article solely reflects the opinions and conclusions of its authors and not TRI or any other Toyota entity. YC is supported by Stanford Graduate Fellowship. TM acknowledges support of Google Faculty Award. The work is also partially supported by SDSI and SAIL at Stanford.

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

## A  PROOFS OF THEOREM 1

We prove a general theorem here. In particular, we have the general theorem below.

**Theorem 2.** *Assume that $f^\star, q, I \in W_2^2$. Suppose (1) $\ell(a, y)$ is convex and three times continuously differentiable with respect to $a$, (2) there exist constants $0 < c < C < \infty$ such that $c \le I(X) \le C$ almost surely, and $\epsilon \triangleq \nabla_a \ell(f^\star(X), Y)$ satisfies $\mathbb{E}[\epsilon|X] = 0$, $\mathbb{E}[\epsilon^2|X] = I(X)$ and $\mathbb{E}[\epsilon^4|X] < \infty$ almost surely. Let $r(t) = -1/(q(t)I(t))$ and $L_0 = \int_{-\infty}^{\infty} \frac{1}{4} \exp(-2|t|) dt$. If we choose $\lambda = C_0 n^{-2/5}$ for some constant $C_0 > 0$, the asymptotic mean squared error of $\hat{f}$ by equation 4 is*

$$\lim_{n \to \infty} MSE(\hat{f}) = C_n \int_0^1 \lambda^2 r^2(t) \left[ \frac{d}{dt} (\rho(t)(f^*)'(t)) \right]^2 + L_0 r(t)^{1/2} \rho(t)^{-1/2} dt$$

*in probability, where $C_n$ is a scalar that only depends on $n$.*

It is easy to check that the logistic loss satisfies the condition of the theorem.

The proof strategy of Theorem 2 is adopted from the proof of Theorem 2 of (Wang et al., 2013) by generalizing it from the least square loss to logistic loss.

The high level idea is to reformulate $\hat{f}$ as solutions to ordinary differential equations. Let $(\gamma_v, h_v)$ be the (normalized) solution of the following equation

$$- \rho(t) h_v''(t) = \gamma_v I(t) q(t) h_v(t), \tag{8}$$

$$h_v'(0) = h_v'(1) = 0, \text{ and } h_v''(0) = h_v''(1) = 0. \tag{9}$$

We define the the leading term of $\hat{f} - f^\star$ as $S_{n,\lambda}(f^\star)$ as

$$S_{n,\lambda}(f^\star) = \frac{1}{n} \epsilon_i K_{X_i} - W_\lambda f^\star, \text{ where}$$

$$K_t(\cdot) = \sum_v \frac{h_v(t)}{1 + \lambda \gamma_v} h_v(\cdot) \text{ and } W_\lambda h_v(\cdot) = \frac{\lambda \gamma_v}{1 + \lambda \gamma_v} h_v(\cdot).$$

By Proposition 2.1 and Theorem 3.4 of Shang et al. (2013), we have

$$\sup_x |\hat{f}(x) - f^\star(x) - S_{n,\lambda}(f^\star)(x)| = o_P(n^{-1/3}). \tag{10}$$

Following the same proof of Theorem 2 of (Wang et al., 2013), we can simplify the definition of $K_t$ and $W^\lambda$ as

$$K_t(x) = \frac{I(t)}{q(t)} J(t, x) \text{ and } W_\lambda f^\star(t) = \lambda r(t) \left[ \frac{d}{dt} (\rho(t)(f^*)'(t)) \right], \tag{11}$$

where $J(t, s) = \beta \rho(s) Q_\beta'(s) L_0 (\beta |Q_\beta(t) - Q_\beta(s)|)$ and $Q_\beta(t, s) = \int_0^t (r(s)\rho(s))^{-1/2} (1 + O(\beta^{-1})) ds$ and $\beta = 1/\sqrt{\lambda}$. Plugging equation 11 into equation 10, we then have

$$\lim_{n \to \infty} MSE(\hat{f}) = C_n \int_0^1 \lambda^2 r^2(t) \left[ \frac{d}{dt} (\rho(t)(f^*)'(t)) \right]^2 + L_0 r(t)^{1/2} \rho(t)^{-1/2} dt$$

## B  IMPLEMENTATION DETAILS

We develop our core algorithm in PyTorch (Paszke et al., 2017).

**Implementation details for CIFAR.**   We follow the simple data augmentation used in (He et al., 2016) with only random crop and horizontal flip. We use ResNet-32 as our base network and repeat all experiments for 3 runs. We use standard SGD with momentum of 0.9, weight decay of $1 \times 10^{-4}$ for training. The model is trained with a batch size of 128 for 120 epochs. We anneal the learning rate by a factor of 10 at 80 and 100 epochs. We group the data by class labels, and by default we split $\mathcal{D}$ equally and randomly into $\mathcal{D}_{\text{train}}$ and $\mathcal{D}_{\text{val}}$. As for the Jacobian regularizer, we sum over the frobenius norm of the gradients of all normalization layers' (BN by default) activations with respect to the classification loss. For experiments of HAR, we tune $\lambda$ so that the largest enforced regularization strength ($\lambda \tau_i$) is 0.1. We train each model with 1 NVIDIA GeForce RTX 2080 Ti.

**Implementation details for IMDB-review.**    We train a two-layer bidirectional LSTM (Huang et al., 2015) with 256 units followed with 0.5 dropout before the linear classifier. The network is trained for 20 epochs with Adam optimizer (Kingma & Ba, 2014). For HAR, we tune $\lambda$ so that the largest enforced regularization strength ($\lambda\tau_i$) is 0.1. We train each model with 1 NVIDIA GeForce RTX 2080 Ti.

**Implementation details for WebVision.**    We use the standard data augmentation same as (He et al., 2016) including random crop and horizontal flip. For mini WebVision, We train the network for 90 epochs using standard SGD with a batch size of 128. The initial learning rate is 0.1 and is annealed by a factor of 10 at epoch 60 and 90. For full WebVision, We train the network for 50 epochs using standard SGD with a batch size of 256. The initial learning rate is 0.1 and is annealed by a factor of 10 at epoch 30 and 40. For experiments of HAR, we tune $\lambda$ so that the largest enforced regularization strength ($\lambda\tau_i$) is 0.1. We train each model with 8 NVIDIA Tesla V100 GPUs.

**Runtime analysis.**    Because the pre-trained model only trains on half of the training data and is only done once, the run-time of HAR is at most twice of the time for ERM. Many baselines in our paper use sophisticated pipelines and are slower than HAR. For example, INCV trains 2 models simultaneously for 4 times from random initialization to get a clean training set. MW-Net has a very slow convergence rate, which is a common issue for meta-learning.

## C  ADDITIONAL RESULTS

### C.1  SIMULATING HETEROSKEDASTIC NOISE ON IMDB-REVIEW.

IMDB-review dataset has a total of 50,000 (25,000 positive and 25,000 negative reviews) movie reviews for binary sentiment classification (Maas et al., 2011). To simulate heteroskedastic noise for this binary classification problem, we project 5% of the labels of negative reviews to positive, and 40% in the reverse direction. Table 6 summarizes the results. The proposed HAR outperforms the ERM baseline with various strength of uniform regularization.

Table 6: Top-1 validation accuracy (averaged over 3 runs) on heteroskedastic IMDB-review dataset.

| Reg Strength | Acc. of neg. reviews | Acc. of pos. reviews | Mean Acc |
|---|---|---|---|
| 0 | $91.9 \pm 2.0$ | $50.9 \pm 1.8$ | $71.4 \pm 0.5$ |
| Unif-reg ($\lambda = 0.01$) | $94.3 \pm 1.8$ | $51.9 \pm 2.0$ | $73.1 \pm 0.3$ |
| Unif-reg ($\lambda = 0.1$) | $91.5 \pm 1.9$ | $64.3 \pm 1.6$ | $77.9 \pm 0.4$ |
| Ours (HAR) | $93.1 \pm 1.5$ | $72.8 \pm 1.7$ | $83.0 \pm 0.3$ |

### C.2  EVALUATION ON CIFAR-100 WITH REAL-WORLD HETEROSKEDASTICITY

It is acknowledged that CIFAR-100 training set contains noisy examples. For instance, some "tiger" examples are labeled as "leopard" ("tiger" is a defined class as well). There are also noisy examples that contain multiple objects, or are more ambiguous in terms of indentity (Song et al., 2020). We find that HAR can improve over uniform regularization on the well-studied CIFAR-100 due to its heteroskedasticity and the results are reported in Table 7.

Table 7: Top-1 validation accuracy (average over 3 runs) of ResNet-32 on the original CIFAR-100.

| Reg Strength | Train Acc | Val Acc |
|---|---|---|
| 0 | $96.0 \pm 0.1$ | $69.8 \pm 0.2$ |
| Unif-reg ($\lambda = 0.001$) | $96.4 \pm 0.1$ | $70.0 \pm 0.2$ |
| Unif-reg ($\lambda = 0.01$) | $95.7 \pm 0.1$ | $70.6 \pm 0.1$ |
| Unif-reg ($\lambda = 0.1$) | $88.8 \pm 0.1$ | $70.5 \pm 0.1$ |
| **Adaptive (HAR)** | $96.2 \pm 0.1$ | $\mathbf{71.4 \pm 0.1}$ |

### C.3 SIMULATING HETEROSKEDASTIC AND IMBALANCED DATASETS ON CIFAR

As mentioned in Section 3.1, we consider another variant of heteroskedastic and imbalanced dataset such that the rare classes have low noise level. To simulate this setting, we make the clean classes have fewer labels than the corrupted classes on the heteroskedastic CIFAR-10 we created in Section **??**.

Table 8 summarizes the results. For the setting of imbalance ratio equals 10, INCV automatically drops 34.1% of examples from the clean and rare classes, which results in a decrease of mean accuracy on the rare and clean classes. HAR is able to achieve improvements on both noisy classes and rare classes by enforcing the optimal regularization.

Table 8: Top-1 validation accuracy (averaged over 3 runs) of ResNet-32 on heteroskedastic and imbalanced CIFAR-10.

| Imbalance ratio | 10 | | 100 | |
| Method | Noisy Cls. | Rare Cls. | Noisy Cls. | Rare Cls. |
|---|---|---|---|---|
| ERM | $59.9 \pm 1.1$ | $65.2 \pm 0.7$ | $60.4 \pm 0.9$ | $7.9 \pm 1.3$ |
| Co-teaching | $63.1 \pm 2.3$ | $59.4 \pm 1.4$ | $53.8 \pm 1.9$ | $4.4 \pm 1.8$ |
| INCV | $74.5 \pm 1.2$ | $63.7 \pm 0.8$ | $68.3 \pm 1.8$ | $2.1 \pm 1.3$ |
| MentorNet | $67.3 \pm 1.6$ | $65.5 \pm 1.2$ | $63.3 \pm 1.5$ | $10.8 \pm 1.9$ |
| L2RW | $65.8 \pm 1.4$ | $66.3 \pm 1.2$ | $62.4 \pm 2.1$ | $11.3 \pm 2.9$ |
| MW-Net | $71.4 \pm 0.6$ | $67.7 \pm 0.6$ | $65.0 \pm 1.6$ | $13.5 \pm 2.4$ |
| GCE | $64.6 \pm 1.1$ | $60.2 \pm 1.3$ | $71.2 \pm 1.9$ | $2.6 \pm 1.4$ |
| DMI | $72.3 \pm 1.5$ | $63.3 \pm 1.2$ | $70.8 \pm 1.7$ | $6.2 \pm 1.9$ |
| **Ours (HAR)** | $\mathbf{76.1 \pm 0.8}$ | $\mathbf{72.1 \pm 1.0}$ | $\mathbf{73.0 \pm 1.6}$ | $\mathbf{26.1 \pm 0.8}$ |

### C.4 COMPARING THE EFFECT OF WEIGHTS ON LOSSES AND REGULARIZERS

As discussed in Section 4, we train several classifiers with alternative weights selection scheme which are not optimal. We consider the following two alternatives. (1) **random**: we draw the weights from a uniform distribution with the same mean as the weights of MW-Net and HAR. (2) **inverse**: we take the inverse of the weights learned by MW-Net and HAR and then normalize them to ensure the average reweighting/regularization strength remains the same.

We conducted experiments on the heteroskedastic CIFAR-10 introduced in Section **??** and the results are summarized in Table 9. We could conclude that changing the weights of the regularizer is a more conservative adaption and less susceptible to uncertain estimation than reweighting.

Table 9: Top-1 validation accuracy (averaged over 3 runs) of ResNet-32 on heteroskedastic CIFAR-10 by changing the weighting scheme.

| Method | Avg. Noisy Cls. | Avg. Clean Cls. |
|---|---|---|
| ERM | $68.6 \pm 0.2$ | $93.6 \pm 0.2$ |
| Reweight (MW-Net) | $75.0 \pm 0.3$ | $94.4 \pm 0.2$ |
| Reweight (random) | $62.1 \pm 0.5$ | $92.9 \pm 0.5$ |
| Reweight (inverse) | $13.1 \pm 0.9$ | $90.9 \pm 0.3$ |
| Unif-reg (optimal) | $75.3 \pm 0.3$ | $94.1 \pm 0.2$ |
| Adap-reg (HAR) | $80.7 \pm 0.3$ | $94.5 \pm 0.2$ |
| Adap-reg (random) | $74.8 \pm 0.4$ | $94.2 \pm 0.2$ |
| Adap-reg (inverse) | $73.2 \pm 0.5$ | $94.0 \pm 0.2$ |

### C.5 VISUALIZATION

In order to better understand how the proposed HAR works on real-world heteroskedastic datasets, we plot the per-class key statistics used by HAR and validation errors in Figure 4. We observe that HAR outperforms the tuned uniform regularization baseline on the majority of the classes.

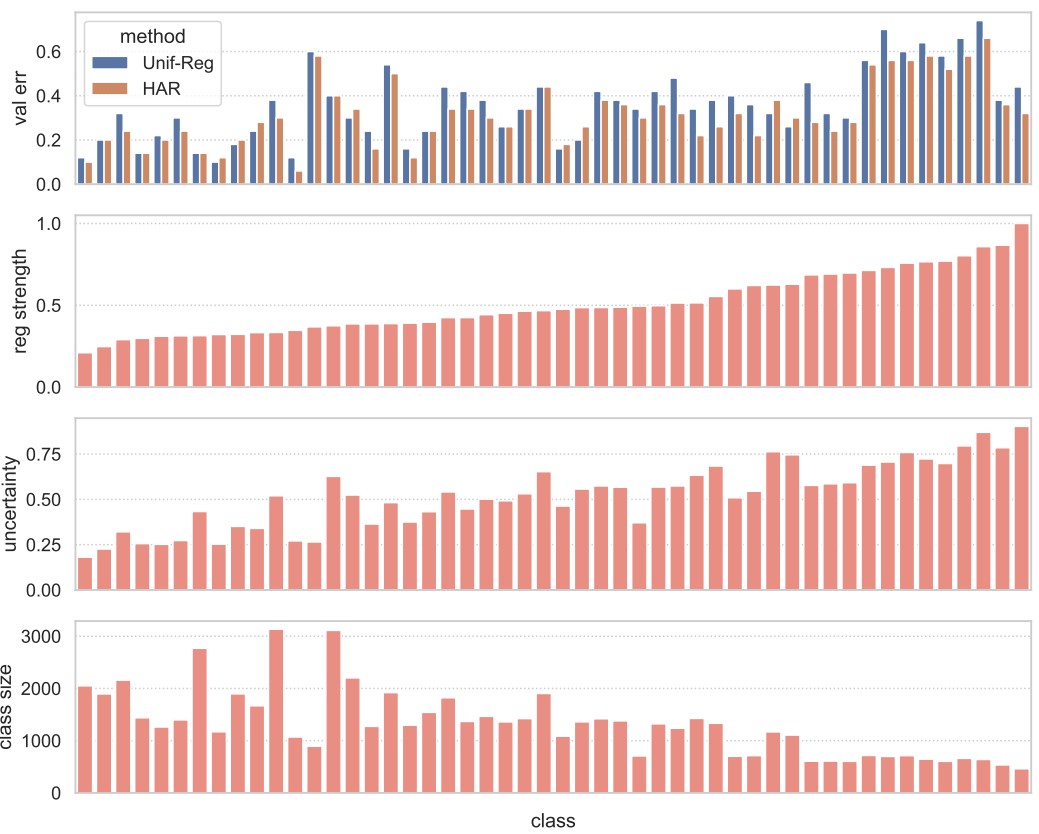

Figure 4: Visualizations of per-class top-1 error and regularization strength of HAR on mini WebVision dataset. The classes are sorted in the ascending order of applied regularization strength.

