# OpenReview forum: "Heteroskedastic and Imbalanced Deep Learning with Adaptive Regularization"
_ICLR.cc/2021/Conference — ICLR 2021 Poster_

### Official Review · AnonReviewer1 · 2020-10-27
**Excellent paper, good improvements**

**Rating:** 9
**Confidence:** 4

**Review:**

The authors propose a novel regularization approach aimed at addressing issues of class imbalance and heteroskedasticity. This adaptive approach uses a Lipschitz regularizer with varying strength in different parts of the input space, regularizing harder in cases of rare and noisy examples. The authors derive the optional regularization strength in the one-dimensional setting, to set ground for the proposed approach and its application in higher-dimensional settings. The approach is evaluated on multiple image datasets, and a textual dataset - and compared to a number of baselines, including those involving noise-cleaning, reweigthing-based methods, meta learning, robust loss functions, as well as tuned uniform regularization. The improvements seem quite strong, and clearly demonstrate the utility of the proposed approach. The paper is well structured, clearly written - and was a pleasure to read.

I only have brief comments:

- Unless I am mistaken (I could have missed it), I didn’t see a concrete mention of a statistical test that was used to determine statistical significance. The authors do use the word ‘significant’ to qualify the observed differences, but that should only be phrased as such if appropriate statistical testing has been done and has been outlined when describing the evaluation. That being said, the improvements are quite stark and I don’t doubt the validity of the claims - I am merely suggesting that the authors should include this information for clarity.

- While the evaluation clearly shows the benefits of the proposed approach, and there is a detailed ablation study, I think that it would be interesting to additionally discuss and analyze in more detail the resulting regularization coming from the proposed approach, on WebVision (or one of the other datasets, wherever it’s easiest) - to show how the distribution of how many examples are being regularized by which factor - and contrast that with the optimal uniform regularization. There could also be a correspondence with Figure 1, if constraining the comparison to freq,rare ; noisy, clean input types. Visualizing some such distributions would help understand the method better - and add to the presented analysis.

---

> ### Author Response · Authors · 2020-11-18
> **Responses to R1**
>
> We thank the reviewer for the positive evaluation of our work. We are glad to hear that the reviewer appreciates the theoretical and empirical contribution, as well as the writing of our paper. Based on the reviewer’s suggestions, we have conducted additional experiments that could further validate our method.
>
> *Q1. “the authors should include appropriate statistical testing.”*
>
> We thank the reviewer for the suggestion. We conducted 5x2cv paired t-test between HAR and the best tuned regularization on heteroskedastic and imbalanced CIFAR-10 and CIFAR-100. The p-values for the above hypothesis testing are 0.005 and 0.009, respectively, confirming the statistical significance of our improvements.
>
> *Q2. “Visualizing some such distributions would help understand the method better.”*
>
> We thank the reviewer for the suggestion. We included a new visualization in Appendix C.7 in the updated pdf. We plot the per-class key statistics used by HAR as well as validation errors and observe that HAR outperforms the tuned uniform regularization baseline on the majority of the classes.

---

### Official Review · AnonReviewer4 · 2020-10-28
**A nice paper**

**Rating:** 7
**Confidence:** 4

**Review:**

**Summary.** This paper presented a novel data-adaptive regularization scheme to adjust for the heteroskedasticity and non-uniformity (i.e., imbalance) of data distribution. Heuristically, solutions are subjected to heavier penalties in regions with either higher aleatoric (large variance) and epistemic (low-sample) uncertainties. While the exact proposal is, in theory, computationally intractable, the author(s) have made some clever relaxations, which lead to a very practical solution.  I very much enjoyed reading this paper.

**Quality & Clarity.** This paper is well organized and clearly written. The. author(s) start the discussion with very intuitive examples,  followed by rigorous mathematical development on how to derive a theoretically justified adaptive penalty. To reach practical solutions, relaxations and surrogates are carefully elaborated. The experiment section is also well-executed, covering convincing synthetic and real-world examples to demonstrate the effectiveness of this proposal, comprehensively compared to SOTA alternatives.

**Originality & Significance.** Although the idea of developing uncertainty-aware models have been repetitively explored in literature, I do find the HAR model proposed in this submission fresh & appealing. One of the main novelty that I appreciate is the fact the author(s) have scaled the gradient penalty for model complexity to deep neural nets, which in the final solution is replaced by the Lipschitz estimate. Although the stratification of sample space feels a bit hacky, I believe that is a necessary compromise to be made. I am reasonably positive this paper is expected to make some impact.

**Minor issues.** There is a missing bracket in the last equation on pp 3. Eqn (6) should perhaps be more explicit on the dependence for $f$. I would love to see discussions, preferably preliminary experiments, to explore the use of nonparametric density estimation schemes to replace sample stratification.

---

> ### Author Response · Authors · 2020-11-18
> **Responses to R4**
>
> We thank the reviewer for the positive evaluation of our work and for appreciating the potential impact of our work, and the theoretical and empirical contribution. We also appreciate the reviewer for confirming the potential impact about our paper.
>
> *Q1. “I would love to see discussions, preferably preliminary experiments, to explore the use of nonparametric density estimation schemes to replace sample stratification.”*
>
> This is an interesting idea. We thank the reviewer for the suggestion. Nonparametric density estimation could indeed potentially allow us to estimate the input density (q) and the aleatoric uncertainty (I) much better. To the best of our knowledge, nonparametric density estimation on high-dimensional noisy data is challenging in practice, so we resorted to a simple assumption of intra-class homogeneity, which we find is reasonable and sufficient in practice. Is there a specific reference on using density estimation for sample stratification that the reviewer has in mind so we could potentially discuss and compare with it in the paper?

---

### Official Review · AnonReviewer2 · 2020-10-28
**A novel task for real-world long-tail learning**

**Rating:** 6
**Confidence:** 4

**Review:**

This paper proposed an adaptive regularization method to handle heteroskedastic and imbalanced datasets, which are closer to real-world large-scale settings. The framework applies a Lipschitz regularizer with varying regularization strength depending on the particular data point. The authors first theoretically study the optimal regularization strength on a one-dimensional binary classification task. By applying some simplification, the result can be extended to high-dimensional multi-class tasks and finally HAR algorithm is proposed. Experiments show that HAR achieves significant improvements over other noise-robust deep learning methods on simulated vision and language datasets with controllable degrees of data noise and data imbalance, as well as a real-world heteroskedastic and imbalanced dataset. The experiments show great improvement. However, since the derivations involve many approximations, the reliability needs to be confirmed by more experiments.

1. Assuming the pre-trained model is sufficiently accurate is not reasonable, especially in your complicated setting.
2. This algorithm divides data into groups according to classes, and assume density and fisher information is constant within each group. This kind of division automatically takes class imbalance into account, which is not necessary according to your theory. Since heteroskedasticity means uncertainty varies between instances, not between classes, what if we divide data into groups with equal size?
3. Synthetic experiments should be conducted to prove at least two things: (1) your algorithm can correctly estimate the regularization strength. (2) your algorithm can be applied to real heteroskedastic datasets, which vary uncertainty between instances, not classes.
4. The imbalanced (long-tail) experiments did not compare with current SOTAs, i.e., BBN (CVPR20).
5. You should compare your method to other methods with similar ideas, i.e., MetaReg.

---

> ### Author Response · Authors · 2020-11-18
> **Responses to R2 (part 2)**
>
> *Q5. “You should compare your method to other methods with similar ideas, i.e., MetaReg.”*
>
> We thank the reviewer for the suggestion. We assume [2] is the paper that [R2] referred to. MetaReg[2] actually proposes to learn a new parametrized regularizer $R(\theta)$ using meta-learning. The regularizer $R(\theta)$ is not data-dependent by their assumption, so all the training examples will have the same regularization strength. Our work is orthogonal to MetaReg[2] since we focus on adaptively selecting the regularization strength. In fact, our proposed HAR algorithm can be potentially combined with any data-dependent regularizer, including a variant of MetaReg[2] if it can be data-dependent.
>
> [1] https://github.com/Megvii-Nanjing/BBN
>
> [2] Balaji, Yogesh, Swami Sankaranarayanan, and Rama Chellappa. "Metareg: Towards domain generalization using meta-regularization." Advances in Neural Information Processing Systems. 2018.

---

> ### Author Response · Authors · 2020-11-18
> **Responses to R2 (part 1)**
>
> We thank the reviewer for the valuable feedback and insightful comments. We have added additional clarifications and experiments based on the reviewer’s comments. We hope the newly provided information could help to strengthen our work.
>
> *Q1.  “The derivations involve many approximations, e.g., assuming the pre-trained model is sufficiently accurate is not reasonable, especially in your complicated setting.”*
>
> We thank the reviewer for this comment. We first derived the optimal regularization strength rigorously in the 1-D setting. As also pointed out by [R4], we believe we made reasonable assumptions to extend the results to higher dimensions but still capture the key facts of the problem --- it is better to apply strong regularization to data with high uncertainty and low density. The estimated q(x) and I(x) will unavoidably contain errors that could not be captured by our theory. The robustness to these errors and the overall effectiveness of our proposed algorithm is validated by empirical experiments [R1, R4].
>
> *Q2. “[the method] assumes density and fisher information is constant within each group. Since heteroskedasticity means uncertainty varies between instances, not between classes, what if we divide data into groups with equal size?”*
>
> We thank the reviewer for the question. If the groups are of equal size, then the regularization strength will only depend on the estimated level of uncertainty. In that case, the regularization strength is still adaptive and Table 4, 7 still show that our method outperforms the uniform regularization strength. However, randomly dividing the data into groups with equal size won’t work because it violates the hypothesis that each group behaves similarly.
>
> A deeper question is what if each instance itself forms a group. By letting the number of groups go to infinity, we can theoretically deal with this setting (with the same conclusion) asymptotically, but practically we won’t have enough samples to estimate the uncertainty and density.
>
> *Q3. “Synthetic experiments should be conducted to prove at least two things: (1) your algorithm can correctly estimate the regularization strength. (2) your algorithm can be applied to real heteroskedastic datasets, which vary uncertainty between instances, not classes.”*
>
> We thank the reviewer for the comment, which we address by the following two points.
>
> (1) We visualized the regularization strength on mini WebVision in the updated pdf to give more intuition on how HAR works on real-world heteroskedastic datasets, as suggested by [R1]. By "can correctly estimate the regularization strength”, we assume the reviewer suggests us to estimate the optimal regularization strength. We would also like to clarify that it is not practical to search the optimal regularization strength because there is one regularization strength for each class and therefore the search space is large.  As a result, we directly show the  empirical effectiveness in improving the test performance over other methods and optimally tuned uniform regularization. In addition, we made another argument in the paper that varying the weights of the regularizer is a more conservative adaptation and less susceptible to inaccurate estimation than re-weighting. We have shown empirical studies to support this claim in Appendix C.6 (Table 9). We hope our second argument could also help to relieve the reviewer’s concern on the correctness of the estimated regularization strength.
> (2) We applied our algorithm to real heteroskedastic datasets with varying uncertainty between instances, e.g., Webvision to demonstrate its effectiveness.
>
> *Q4. “The imbalanced (long-tail) experiments did not compare with current SOTAs, i.e., BBN (CVPR20).”*
>
> We thank the reviewer for the suggestion. We add comparisons with BBN on imbalanced CIFAR. We copied the results reported in the original paper for the long-tailed setting and reproduced results for the step imbalance setting using the official implementation [1]. It could be concluded from the table that HAR outperforms BBN in a total of 7 / 8 test settings. We also included this additional baseline in the updated pdf.
>
> | Dataset | CIFAR-10 LT 100 | CIFAR-10 LT 10 | CIFAR-10 ST 100 | CIFAR-10 ST 10 | CIFAR-100 LT 100 | CIFAR-100 LT 10 | CIFAR-100 ST 100 | CIFAR-100 ST 10 |
> |---------|-----------------|----------------|-----------------|----------------|------------------|-----------------|------------------|-----------------|
> | BBN     | 20.18           | 11.68          | 21.64           | 11.99          | 57.44            | 40.88           | 57.44            | 40.36           |
> | HAR     | 20.46           | 10.62          | 20.27           | 11.58          | 55.35            | 38.98           | 51.73            | 37.54           |

---

### Official Review · AnonReviewer3 · 2020-10-29
**Paper 926 Review**

**Rating:** 5
**Confidence:** 3

**Review:**

The authors propose a regularization approach for heteroskedastic data that regularizes in a data-dependent fashion, so different regions of the data spaces may be regularized differently by preferentially targeting high uncertainty and low density regions. This is achieved by estimating the noise level and density of each training example and then optimizing a regularized objective with input-dependent regularization.

How is k selected in (5)? Also, what is the purpose of Section 2 if neither the one-dimensional case nor the nonparametric model will be used in the experiments. The whole section seems distracting from the main message of the paper.

What is the definition of rare classes in the experiment in Table 1? it seems accuracy for the rare and noisy classes is calculated on all examples, including the 40% for which labels were exchanged. It would be interesting to see the accuracy also calculated for the correct (non-exchanged) labels, is that Table 4?.

Why are the test accuracies in Table 3 better than the validation accuracies in Table 2?

The results in Table 3 are certainly impressive, however, additional results in the Appendix are less so. Particularly, the experiment where HAR is compared to uniform regularization. It would be interesting to see similar results for WebVision. Also, why only exchange 5% of the negative samples but 40% of the positive labels in the IMDB experiment?

Minor:
- $f$, $l$ and $\lambda$ not defined in (1).
- $P(Y=y|X=x)$ in (2) should not be a function of $y$.
- why write the results for HAR and clean classes in bold?
- If the expectation in (4) is explicitly over the dataset why not simply write is as an average?

---

> ### Author Response · Authors · 2020-11-18
> **Responses to R3**
>
> We thank the reviewer for the valuable comments. We respectfully ask the reviewer to consider increasing the score if our clariﬁcation has addressed the concerns raised by the reviewer.
>
> *Q1. “What is the purpose of Section 2?”*
>
> We thank the reviewer for the question. In section 2 we first derive the regularization strength depending on the uncertainty and density of the samples through a rigorous theoretical analysis. We then introduce the proposed HAR algorithm by carefully illustrating the assumptions and approximations we made when extending the 1-D results to high-dimensions. We hope the way we present Section 2 could help to shed light on the principle and intuition of the proposed algorithm, beyond just its practical effectiveness, which we demonstrate on synthetic and real-world datasets.
>
> *Q2. “What is the definition of rare classes in the experiment in Table 1? it seems accuracy for the rare and noisy classes is calculated on all examples, including the 40% for which labels were exchanged. It would be interesting to see the accuracy also calculated for the correct (non-exchanged) labels, is that Table 4?.”*
>
> We thank the reviewer for the question. For the setting in Table 1, we create two types of classes, (a) noisy and rare classes, which have exchanged 40% labels and also got downsampled, and (b) clean classes.
> We report the mean accuracy on the validation set, where all the examples have correct (and non-exchanged) labels.
>
> *Q3. “Why are the test accuracies in Table 3 better than the validation accuracies in Table 2?”*
>
> We thank the reviewer for the question. We use InceptionResNet-v2 for the results in Table 3 and use ResNet-50 as backbones for the results in Table 2. We vary the architecture to demonstrate that the performance improvement is consistent across architectures.
>
> *Q4. “Particularly, the experiment where HAR is compared to uniform regularization. It would be interesting to see similar results for WebVision.“*
>
> We thank the reviewer for the question. The comparison with uniform regularization for WebVision is in Table 2.
>
> *Q5. “Also, why only exchange 5% of the negative samples but 40% of the positive labels in the IMDB experiment?”*
>
> We thank the reviewer for the question. We want to simulate heteroskedastic noise for this binary classification problem, so we change different ratios of labels to the opposite for positive and negative classes, so that the noise level for the two classes are different
>
> *Q6. “How is k selected in (5)?”*
>
> We are not sure we understand the question, as there is no k in Eq.5. Could the reviewer please clarify?

---

### Decision · Program_Chairs · 2021-01-07
**Final Decision**

**Decision:**

Accept (Poster)

**Comment:**

This paper got 2 clear acceptance and 2 borderline recommendation. The main concerns lie in the clarity of the experiment results and settings (AR3). The authors address these questions in their response. AR2 has two important questions. One is whether the simplified assumption holds in the considered very complicated settings (i.e., the labels are noisy and long-tailed). The other one is the lack of comparison with SOTA method for long-tailed classification. The authors did good job in their response. They provide additional experiment results to address these questions. Overall, the quality of this submission meet the bar of ICLR acceptance, though AC has concerns on the complicated settings and the marginal performance improvement over the existing long-tailed works.